ecology, palaeontology, evolution

faunal provincialism, endemicity, Laramidia, sampling bias, Ceratopsidae, Hadrosauridae

**Author for correspondence:**
Susannah C. R. Maidment
e-mail: susannah.maidment@nhm.ac.uk

# Deep-time biodiversity patterns and the dinosaurian fossil record of the Late Cretaceous Western Interior, North America

Susannah C. R. Maidment[1,3], Christopher D. Dean[1], Robert I. Mansergh[1,2] and Richard J. Butler[3]

[1]Department of Earth Sciences, Natural History Museum, Cromwell Road, London, SW7 5BD, UK
[2]Department of Earth Sciences, 5 Gower Place, London WC1E 6BS, UK
[3]School of Geography, Earth and Environmental Sciences, University of Birmingham, Edgbaston, Birmingham B15 2TT, UK

SCRM, 0000-0002-7741-2500; RJB, 0000-0003-2136-7541

In order for palaeontological data to be informative to ecologists seeking to understand the causes of today's diversity patterns, palaeontologists must demonstrate that actual biodiversity patterns are preserved in our reconstructions of past ecosystems. During the Late Cretaceous, North America was divided into two landmasses, Laramidia and Appalachia. Previous work has suggested strong faunal provinciality on Laramidia at this time, but these arguments are almost entirely qualitative. We quantitatively investigated faunal provinciality in ceratopsid and hadrosaurid dinosaurs using a biogeographic network approach and investigated sampling biases by examining correlations between dinosaur occurrences and collections. We carried out a model-fitting approach using generalized least-squares regression to investigate the sources of sampling bias we identified. We find that while the raw data strongly support faunal provinciality, this result is driven by sampling bias. The data quality of ceratopsids and hadrosaurids is currently too poor to enable fair tests of provincialism, even in this intensively sampled region, which probably represents the best-known Late Cretaceous terrestrial ecosystem on Earth. To accurately reconstruct biodiversity patterns in deep time, future work should focus on smaller scale, higher resolution case studies in which the effects of sampling bias can be better controlled.

## 1. Introduction

In order to predict how biodiversity patterns on today's Earth will respond to climate change, the factors that cause biodiversity distributions must be understood [1,2]. Deep-time perspectives can provide novel insights into the controls on biodiversity distribution. By examining biodiversity distributions at times in Earth's history when climate, continental arrangement, and oceanic currents were different than today, ecological hypotheses about the causative mechanisms behind biodiversity distribution and the establishment of modern patterns can be tested [3–7]. However, if palaeontologists wish their data to be informative to those working on the causative mechanisms of modern-day biodiversity patterns, we must first demonstrate that actual biodiversity patterns are preserved in our reconstructions of past ecosystems, and that we are able to overcome the many sampling biases that affect the fossil record (e.g. [8–10]).

Dinosaurs are an exceptional model system for studying biodiversity and macroevolution in terrestrial vertebrates. For greater than 150 million years, from the Late Triassic to the end of the Cretaceous, they dominated terrestrial ecosystems, occupied every continent, and radiated into a wide variety of

ecological niches. Because of public interest, they are the best-sampled Mesozoic terrestrial vertebrate group and their fossils have been collected for well over 150 years [11–14]. Arguably, the best-sampled part of the dinosaurian fossil record is the Late Cretaceous of the Western Interior region of North America [15–17]. During the Late Cretaceous, North America was divided into two landmasses, Laramidia to the west and Appalachia to the east, by the epicontinental Western Interior Seaway. In the latter stages of the Late Cretaceous (Campanian and Maastrichtian), large-bodied herbivorous niches in Laramidia were dominated by two groups of ornithischian dinosaurs, the hadrosaurs and ceratopsids. The body fossil record of the latter is entirely restricted to North America at this time, with the exception of a single taxon [18,19]. Study of these herbivorous dinosaurs has provided major insight into dinosaur behaviour, palaeoecology, and biogeographic patterns (e.g. [20–32]).

Numerous workers have argued for strong faunal provinciality in Laramidia throughout the Campanian and have divided the landmass into northern and southern faunal provinces (e.g. [33–35]). This signal is particularly clear in chasmosaurine ceratopsids, where virtually all species are recognized from either northern or southern Laramidia, but not both [34–36]. This endemicity in ceratopsids is thought to have driven high levels of diversity, underpinning their radiation [34]. Since no geological or geographical barrier has thus far been identified between northern and southern Laramidia [37], the boundary between the northern and southern provinces has been suggested to be related to latitudinal climate, temperature, or rainfall patterns [34,35,38] or was maintained due to competition between local populations [39]. The patterns of apparent provincialism decrease in the Maastrichtian, coincident with overall regression of the Western Interior Seaway [15,17,33,36].

These hypotheses of biogeographic provincialism, however, remain controversial. With very few exceptions [35,36], studies that advocate for provincialism are based on qualitative observations (e.g. [33,34]) and arise from comparisons of the fauna of specific geological formations (e.g. [33–35]). Recent research has, however, suggested that some of the formations used in such studies are not contemporaneous [17,40] and that the length of time intervals used results in the amalgamation of multiple successive faunas [17,37]. Many studies advocating faunal endemism are based on taxonomic decisions that have proven controversial and the conclusions have been called into question as a result (e.g. [37,39]). Additionally, it remains a possibility that faunal provinces within the Campanian are an artefact of sampling: most Campanian dinosaur occurrences are known from Alberta, Montana, southern Utah, and northern New Mexico, with far less sampling having occurred in northern Utah and Wyoming [16].

Three quantitative studies have investigated the provincialism hypothesis in dinosaurs of the Late Cretaceous Western Interior. Gates *et al.* [35] used a variety of statistical techniques to assess the similarity between Campanian northern and southern faunas, and found evidence for either two distinct provinces with a broad area of overlap between them, or a latitudinal diversity gradient. The statistical techniques employed were unable to distinguish between these two hypotheses, and their results regarding dinosaurs were inconclusive. They suggested a further investigation into the causes of dinosaur distribution in the Western Interior. Berry [36] used a phylogenetic approach to assess biogeography within the Campanian and found no evidence for endemic sub-clades of chasmosaurine ceratopsids, arguing that this would be expected if there was a major barrier to dispersal, or niche conservatism related to climate. Vavrek & Larsson [15] investigated faunal endemism in the Maastrichtian of Laramidia using measures of beta diversity. After correcting for sampling biases, they found little evidence of provincialism, instead suggesting a homogeneous dinosaurian fauna across the Western Interior region at the very end of the Cretaceous; however, they did not test to see whether apparent biogeographic patterns within the Campanian were also caused by sampling.

Herein, we quantitatively test hypotheses of faunal endemism in both the Campanian and Maastrichtian using biogeographic and multivariate statistical approaches. We focus our study on ceratopsid and hadrosaurian dinosaurs, as these megaherbivores have well-understood phylogenies and taxonomies and have been at the centre of previous discussions of faunal provinciality in this region. The distinctiveness of northern and southern Laramidian provinces are tested using phylogenetically corrected Biogeographic Connectedness (pBC). This quantitative method uses a network approach to assess phylogenetic distances between taxa in different geographic areas, resulting in a metric that quantifies the degree of faunal provinciality versus cosmopolitanism. It has been used successfully to understand changes in faunal compositions through the Carboniferous–Permian transition [41], and the Permian–Triassic and Triassic–Jurassic extinction events [42]. We also introduce additions to the methodology that address concerns regarding variation in sampling through time. To investigate the impact of sampling bias on our results, we examine correlations between occurrences (records of specimens) and collections (sites where specimens have been collected) with latitude, and use multivariate regression to examine which sources of sampling bias best explain sampling patterns. We use our results to determine whether it is possible to identify true geographic patterns of biodiversity on a continental scale in this very well-sampled area.

## 2. Methods

### (a) Taxon sampling and phylogeny

Since no complete phylogenetic analysis of all ceratopsids is available, we built an informal supertree of all ceratopsid taxa considered valid in recent phylogenetic analyses by combining the phylogenetic results of [18] for chasmosaurines and [19] for centrosaurines. We resolved polytomies by removing *Nedoceratops*, a taxon some workers consider to be invalid ([43], but see [44]), from the data matrix in [18] and re-analysing the dataset. This resolved polytomies in the clade containing the common ancestor of *Eotriceratops*, *Triceratops*, and all of its descendants. The resulting supertree includes 67 taxa and represents a consensus of current views on ceratopsian phylogeny (electronic supplementary material, figure S1a).

The structure of the hadrosaurid tree is based on several key recent analyses [45–47]. We resolved polytomies and added taxa considered valid but not included in those references using other recent phylogenetic analyses [48–51]. The resulting supertree includes 55 taxa and represents a current reasonable estimate of hadrosaur phylogeny (electronic supplementary material, figure S1b).

## (b) Stratigraphic age and geographic data

Age for North American hadrosaur and ceratopsid species was obtained from the primary literature. The formations in which taxa occurred were found from the Paleobiology Database (PBDB; www.paleobiodb.org), and the most recent absolute age estimate of those formations was obtained from the primary literature (see electronic supplementary material, OSM, for sources). The age and geographic data for taxa outside of North America were obtained from the PBDB. pBC requires *a priori* assignment of geographic regions to test hypotheses of biogeographic connectedness, so we assigned dinosaurs to either northern Laramidia or southern Laramidia. Northern Laramidia includes taxa found in Wyoming and further north; southern Laramidia includes taxa found in Utah and further south, following previous studies. Age data were used to time-calibrate the phylogenetic trees using the 'timePaleoPhy' function of the Strap package [52] in R v. 3.5.2 [53] with the minimum branch length option specified (type='mbl'). While it would be ideal to use high-resolution bins to test patterns of biogeography through the Late Cretaceous [17], too few taxa would be present in each bin to permit the use of pBC. Consequently, we divided taxa into Campanian and Maastrichtian time bins, which also has the benefit of allowing for comparison between previous studies of faunal provincialism in this area. Where a taxon's stratigraphic range/uncertainty crossed the Campanian–Maastrichtian boundary, it was included in both time intervals.

## (c) pBC

We calculated pBC for Campanian ceratopsids, Maastrichtian ceratopsids, and Campanian hadrosaurs. The sampling of Maastrichtian hadrosaurs was too sparse, particularly in southern Laramidia, to calculate meaningful pBC values. Trees were pruned to exclude taxa from timeslices other than the one being analysed, and were made ultrametric prior to analysis. pBC was calculated using the function BC of the package 'dispeRse' (available at github.com/laurasoul/disperse). We initially varied the constant $\mu$ (see [41]) from 1 to 15 million years; subsequent analyses used a constant $\mu$ of 10. Data were jack-knifed 1000 times to produce a distribution of possible pBC values. To address concerns about the potential for a relationship between pBC and taxon sample size [54], we calculated rarefaction curves for pBC for the ceratopsian data (to facilitate comparisons between the Campanian and Maastrichtian). Sample sizes were rarefied down to a minimum number of five taxa. Ninety-five per cent confidence intervals for the rarefaction curves were generated using 1000 replicates at each sampling level.

## (d) Randomization of data (null model)

In order to determine whether pBC for each clade and time interval was significantly different from random, we randomly permuted the geographic areas in which taxa are found. We generated 1000 permutations of the data for each clade and time interval and calculated pBC for each permutation. The pBC for the unpermuted data was compared to the distribution of permuted pBC values to establish statistical significance ($p < 0.05$).

## (e) Sampling bias

To investigate whether biogeographic patterns we observed in the pBC results were influenced by sampling bias, we downloaded raw occurrence data for ceratopsids and hadrosaurs for the Campanian and Maastrichtian from the PBDB. We then downloaded North American dinosaur-bearing collections and North American tetrapod-bearing collections for each timeslice,

and plotted occurrences and collections with latitude. We compared the curves using Spearman's rho and Kendall's tau.

To investigate the possible causes of sampling bias we identified, we statistically examined correlations between occurrences and outcrop area, depositional environment, and proxies for exposure. First, we imported publicly available United States Geological Survey (USGS) state-level and Canadian Province digital geological maps (www.ngmbd.usgs.gov; https://ags.aer.ca/publication/map-600; https://geohub.saskatchewan.ca/datasets/bedrock-geology) into ArcMap 10 (www.esri.com), identified Campanian and Maastrichtian strata, and assigned an environmental attribute determining whether strata were deposited in a terrestrial, marine, or mixed setting (OSM). These data, along with maximum green vegetation fraction (MGVF) and slope, both proxies for exposure, were imported into R (version 3.5.0). Methods for generating MGVF and slope are provided in OSM. Level plots of total outcrop area, terrestrial, mixed and marine outcrop area, slope, and MGVF were produced using the levelplot() function of the rasterVis() package [55] (electronic supplementary material, figure S2).

To investigate the power of each or a combination of these variables to explain the dinosaur occurrence data, we carried out a model-fitting approach using generalized least-squares regression (GLS). Ceratopsian and hadrosaur occurrences from the PBDB were counted in each 1-degree latitudinal bin (latitude is modern latitude, rather than palaeolatitude). Models compared latitudinal changes in ceratopsian and hadrosaur occurrences to changes in four different measures of outcrop area (see OSM), MGVF, and slope. GLS autoregressive models were fitted to combinations of potential explanatory variables. We used a first-order autoregressive model (corARMA) fitted to the data to account for spatial autocorrelation using the function gls() in the R package nlme v. 3.1–150 [56]. GLS reduces the chance of overestimating the statistical significance of regression lines due to serial correlation in the latitudinal series.

Data series were ln-transformed prior to analysis to ensure normality and homoskedasticity of residuals. We calculated likelihood-ratio-based pseudo-$R^2$ values using the function r.squaredLR() of the R package MuMIn [57]. Results were compared using Akaike's information criterion for small sample sizes (AICc) and Akaike weights were calculated to identify the best combination of explanatory variables from those tested. AICc was calculated using the function AICc() of the R package qpcR [58], and Akaike weights calculated using the aic.w() function of the R package phytools [59].

## (i) Sampling bias and pBCs

To test the impact of the Campanian bimodal sampling distribution on pBC results, we ran a second pBC test where we randomly removed 95% of ceratopsian taxa from the Maastrichtian that occurred between 35 and 50 degrees of latitude. We chose these latitudinal boundaries to enforce a similar bimodal latitudinal diversity gradient on the Maastrichtian data as seen in the Campanian (see Results). The remaining distribution of occurrences was used to re-run pBC analyses (with a $\mu$ of 10), and this process was repeated 1000 times for increased accuracy of results. pBC scores were recorded for each run, and the resulting distribution was used to calculate the mean pBC to compare against the original Maastrichtian ceratopsian pBC score and produce a probability density curve to estimate the probability of different values of pBC scores.

# 3. Results

The observed value of pBC for Campanian ceratopsids was 0.05, while that for Campanian hadrosaurs was 0.11, and for Maastrichtian ceratopsids the observed value was 0.16.

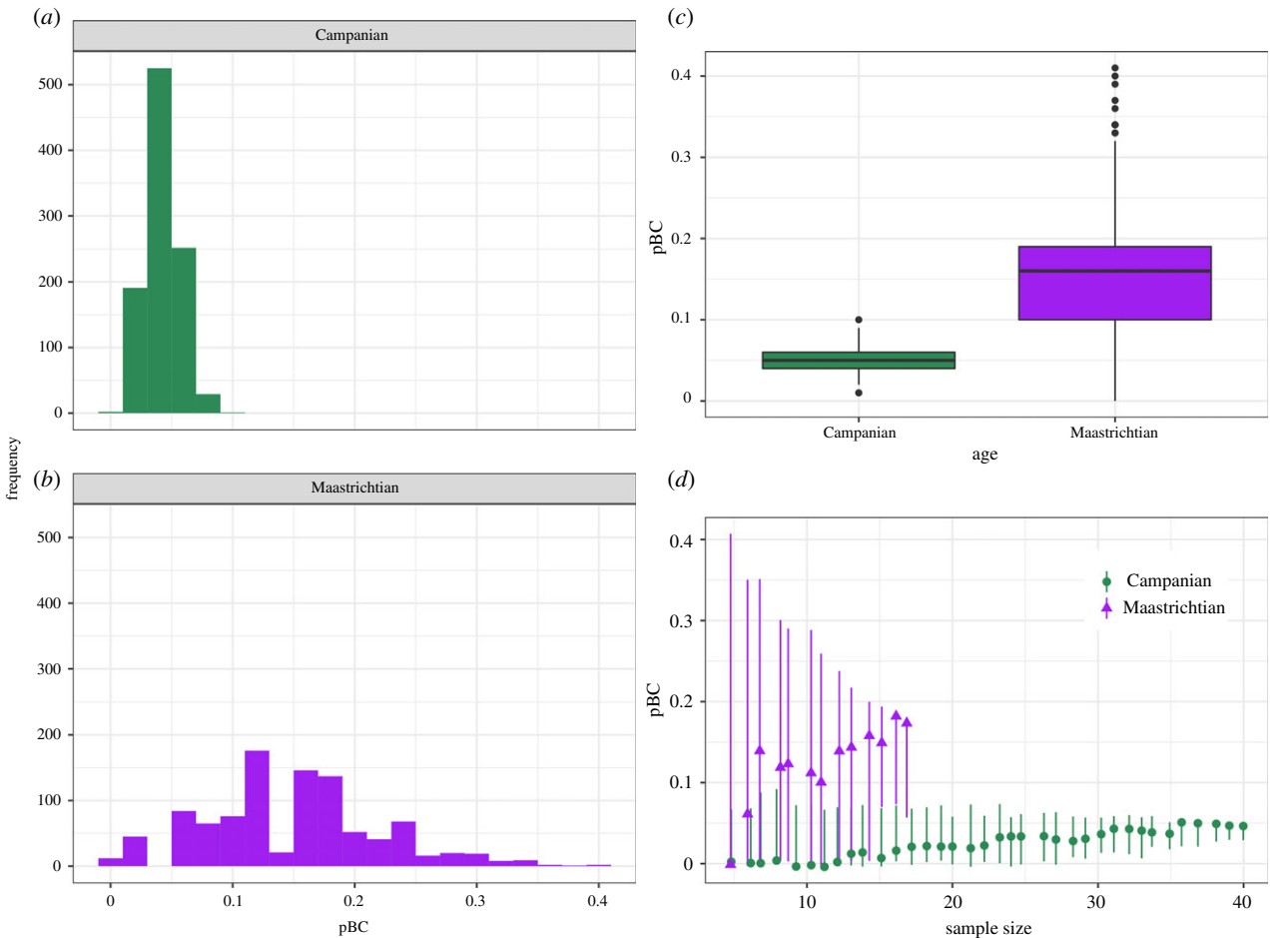

**Figure 1.** (*a–c*) Jack-knifed distributions of pBC values for Campanian (green) and Maastrichtian (purple) ceratopsids. (*d*) Rarefaction curves for Campanian (green circles) and Maastrichtian (purple triangles) ceratopsids. Error bars show the 95% confidence intervals of values obtained during rarefaction. (Online version in colour.)

pBC was therefore lower for ceratopsids in the Campanian than in the Maastrichtian, and endemism was correspondingly higher, in agreement with previous studies [33,36]. Jack-knifed distributions of ceratopsid pBC for the Campanian and Maastrichtian overlap (figure 1*a–c*), but their median values are strongly significantly different from each other (Wilcox Test, $W = 60235$, $p = 0.00$). Rarefaction curves for ceratopsids for the Campanian and Maastrichtian indicate a much higher pBC in the Maastrichtian than in the Campanian at equivalent levels of sampling, although the confidence intervals do overlap, particularly at lower sampling levels (figure 1*d*). This demonstrates that the higher pBC of the Maastrichtian is not a consequence of sampling lower numbers of species in that interval in comparison to the Campanian. Higher pBC equates to more cosmopolitan faunas, and thus this result supports lower endemism in Laramidia during the Maastrichtian when compared to the Campanian.

Values for pBC for both Campanian and Maastrichtian ceratopsid data are statistically significantly lower than for datasets in which the geographic areas are randomized (Campanian, $p = 0.00$; Maastrichtian, $p = 0.015$; electronic supplementary material, figure S3), and the same is true for the Campanian hadrosaur data ($p = 0.00$; electronic supplementary material, figure S3). This indicates that endemism was statistically significantly higher than in all randomized datasets across both time intervals and supports previous qualitative hypotheses of distinct northern and southern provinces in Laramidia (e.g. [33–35])

Curves of raw occurrence data with latitude for hadrosaurs and ceratopsids in both the Campanian and the Maastrichtian correlate strongly and statistically significantly with both dinosaur-bearing and tetrapod-bearing collections (figure 2; electronic supplementary material, figure S4; OSM). During the Campanian, sampling and occurrences are focused at two latitudes: 51–49 degrees north, which corresponds with the Dinosaur Park, Oldman and, to a lesser extent, the Foremost formations, and 36–37 degrees north, which corresponds primarily with the Kirtland/Fruitland, Aguja, and Kaiparowits Formations (figure 2*a,b*; electronic supplementary material, figure S4*a,b*). These two areas have been sampled orders of magnitude better than the surrounding latitudinal bins [16], although there are tetrapod- and dinosaur-bearing formations across the majority of the Western Interior at this time (figure 2*a,b*; electronic supplementary material, figure S4*a,b*). In the Maastrichtian, sampling is more evenly spread across the range of latitudes for which we have hadrosaur and ceratopsid body fossils (figure 2*c,d*; electronic supplementary material, figure S4*c,d*; [16]). These data are strongly indicative that the provinciality observed based on raw data in the Campanian could be due to intensive sampling in the Dinosaur Park Formation and Kirtland/Fruitland Formations with a lack of sufficient sampling between, and our observed increase in pBC (= reduced endemism) in the Maastrichtian is due to increased latitudinal coverage of sampling.

The mean pBC score of Maastrichtian ceratopsians subjected to a Campanian-style sampling distribution was

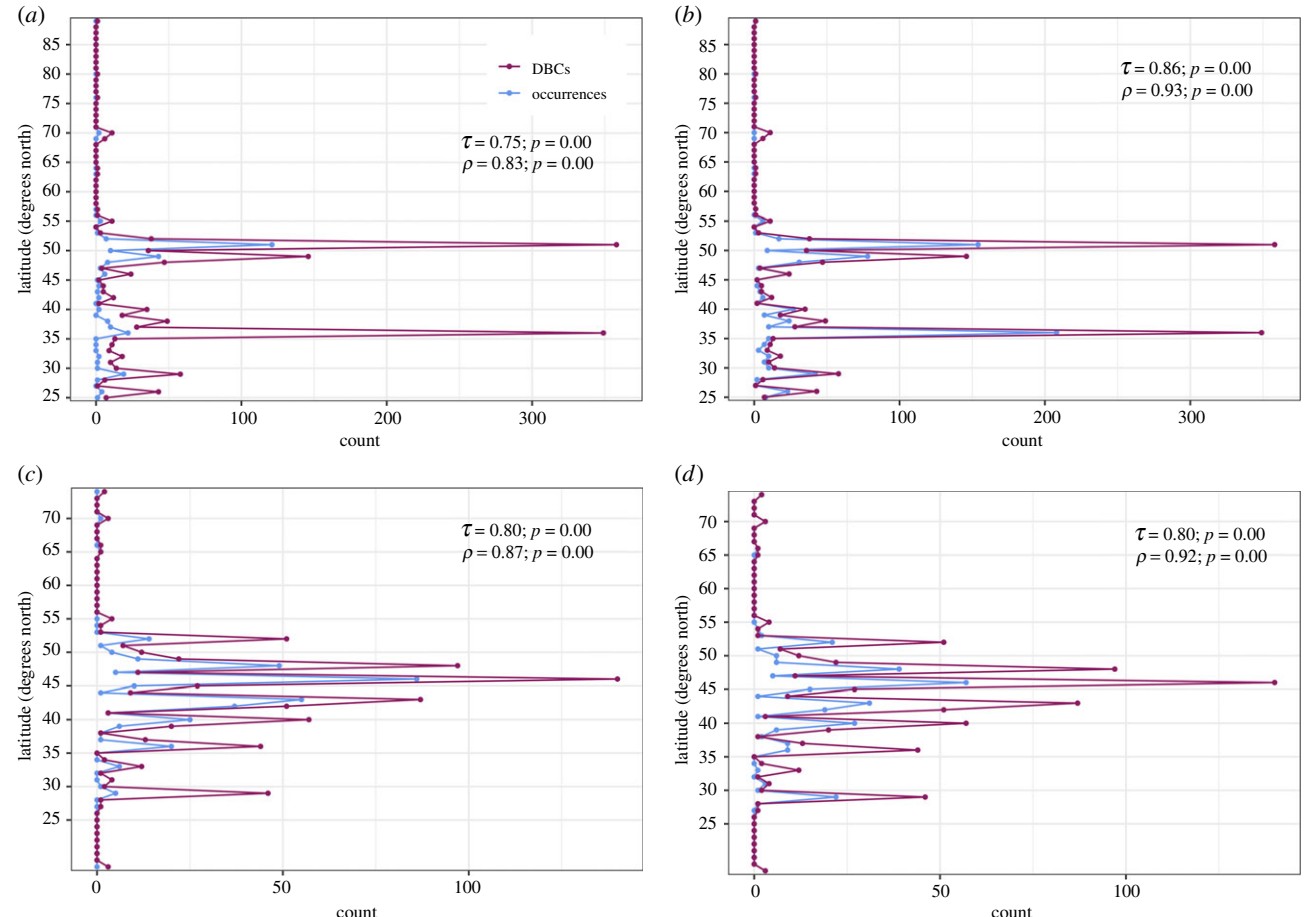

**Figure 2.** (a,c) Ceratopsid occurrences and dinosaur-bearing collections with latitude in the (a), Campanian and (c), Maastrichtian. (b,d) Hadrosaurid occurrences and dinosaur-bearing collections with latitude in the (b), Campanian and (d), Maastrichtian. $\tau$ = Kendall's tau; $\rho$ = Spearman's rho; DBCs, dinosaur-bearing collections. (Online version in colour.)

0.0351 with a standard deviation of 0.0476, significantly lower than the original pBC score of 0.16. The probability of a pBC score less than or equal to 0.8 was 0.78 (OSM and electronic supplementary material, figure S5). These results provide a further indication that sampling bias is driving pBC scores of Campanian fauna.

A lack of sampling in the area between 49 degrees north and 37 degrees north (the 'sampling peaks') in Campanian strata could be caused by a variety of factors. It has long been known that rock outcrop area is strongly correlated with raw diversity (e.g. [60,61]); if there is less outcrop, there are fewer opportunities for palaeontologists to sample the rocks, and fewer fossils found as a consequence. As terrestrial organisms, the vast majority of dinosaur fossils are found in formations that were deposited on land. If Campanian rocks between the sampling peaks are primarily marine, there will be fewer opportunities for dinosaur fossils to be preserved, and thus fewer opportunities for them to be sampled by palaeontologists. Fossils are primarily found where bare rock is exposed at the surface. If less rock is exposed between the sampling peaks than in the areas of the peaks themselves, there will be fewer opportunities for fossils to come to light.

GLS analyses recovered the following best models (highest AICc weights) for outcrop and tetrapod occurrence masks (see OSM for additional results): Campanian hadrosaurs, summed outcrop area + MGVF + slope; Campanian ceratopsians, non-marine total outcrop area; Maastrichtian hadrosaurs, null model; Maastrichtian ceratopsians, null model. However, in

nearly all cases the correlations are non-significant (OSM) and only the Campanian hadrosaur model results had a strong overall explanatory power (OSM). This indicates that the potential sampling bias with latitude in the Campanian that we have identified cannot be fully explained by any of these variables and other sources of sampling bias that are hard to quantify may additionally be responsible.

## 4. Discussion

Several authors have suggested that the apparent faunal provincialism in Laramidia during the Late Cretaceous is an artefact, either because the formations in which dinosaurian taxa have been found are not contemporaneous [17,37,40] or due to uneven sampling of the fossil record [15,16]. Our results show that while the raw data clearly supports faunal endemicity, particularly in the Campanian, this pattern is driven by a lack of sampling outside of two specific latitudinal belts on Laramidia (51–49 degrees north, which corresponds with the Dinosaur Park, Oldman and Foremost formations, and 36–37 degrees north, which corresponds primarily with the Kirtland/Fruitland, Aguja, and Kaiparowits formations). This sampling bias cannot be fully explained by differences in outcrop area across the region, or by differences in slope or vegetation, which are factors that affect rock exposure.

There are numerous other factors that can bias sampling, but these are very difficult to quantify. Low sampling

between the northern and southern sampling peaks could occur if palaeontologists have yet to prospect the area to the same degree that they have in the north and south. The Late Cretaceous of the Western Interior has been intensively sampled for dinosaur fossils for over 100 years, and it is now probably the best-known Late Cretaceous ecosystem anywhere on Earth [15,16]. It is therefore highly unlikely that large parts of it remain unexplored for dinosaurs, and the fact that dinosaur fossils are known from the area between the sampling peaks during the Maastrichtian suggest the area has been explored. The lack of exploration for fossils is therefore unlikely to be the primary driver of the uneven sampling patterns we have observed.

The 'common cause hypothesis' (e.g. [62]) suggests that correlations between raw diversity and sampling proxies (e.g. numbers of formations) are driven by a third factor, usually sea level. Although initially formulated for marine environments, the possibility of a sea-level driven common cause on land has also been discussed (e.g. [63]). During sea-level high stands sediment flux to inner shelves and marginal marine areas is high; this results in both high potential for the preservation of fossils due to rapid burial and high diversity due to habitat fragmentation leading to endemism and increased beta diversity. Conversely, sediment bypasses inner shelf environments during low stands, reducing sediment flux and leading to poorer preservation of fossils due to a lower chance of rapid burial, while diversity is lower due to cosmopolitanism. Although the effect of eustatic sea-level changes on the global terrestrial fossil record of vertebrates has been questioned [63], Chiarenza et al. [16] demonstrated that the areas of our northern and southern sampling peaks correlated with high sediment fluxes and low runoff rates during the Campanian. It is therefore possible that reduced sampling between our sampling peaks is because this area was less suitable for fossil preservation in the Campanian. Indeed, Chiarenza et al. [16] suggested that faunal provinicialism in the Campanian was a sampling bias at least partially due to variation in climatic-induced taphonomic suitability between northern and southern regions.

Historical collecting practises and/or land ownership might also play a role in the sampling patterns we have observed. If the proportion of outcrop on public land was reduced in the areas outside of our sampling peaks, this might mean palaeontologists have less access to explore there for fossils. Furthermore, if there is a particularly field-active palaeontological institution close to an area of Campanian outcrop, or long-term agreements in place with landowners, this may have allowed prospecting to occur more regularly over a longer period of time in specific areas. A bias may also be introduced by uneven regional entry of data into online databases such as the PBDB. Such a bias could stem from monographs on specific formations or museums whose collections focus on specific areas that also have online databases. Data from these sources are comparatively easy to enter into the PBDB and thus could be contributing to the sampling patterns we observed.

It seems highly likely that a combination of available outcrop area, rocks suitable for the fossilization of vertebrate remains, and an interplay between climate, topography, and historical collection and data entry practises is responsible for variations in sampling across the Western Interior,

which have resulted in apparent northern and southern faunal provinces on Laramidia.

## (a) Taxonomic differences in northern and southern Laramidia

Despite the fact that we find faunal provincialism in the Late Cretaceous Western Interior to mostly likely be due to sampling bias based on currently available data, it is clear that different taxa are found in the northern and southern areas of Laramidia [33–35]. This is especially clear in chasmosaurine ceratopsids, where there is almost no overlap at all between taxa found in the north and those found in the south [34], but see [36,39]. It has been demonstrated that many of these taxa were not contemporaneous [40], which would at least partially explain taxonomic differences. But, in addition, the study area covers 12 degrees of latitude and climate would have varied significantly over that area, even in a greenhouse world where latitudinal temperature gradients were reduced relative to today [39,64]. General circulation models for the Campanian show significant variation in mean annual temperature and rainfall patterns with latitude across Laramidia [16] and recent research has suggested elevated temperature gradients in a climatic transition zone between the northern and southern faunal provinces [38]. Given that there is evidence for both spatial [26] and functional [29] niche partitioning in Laramidia's large herbivores, taxonomic differences between the north and south could be related to climatic preference, and there may well have been a latitudinal biodiversity gradient across the area. Unfortunately, we have demonstrated here that that raw data is currently too influenced by sampling biases for such biodiversity patterns to be reconstructed.

## 5. Conclusion

We show that data quality of Campanian and Maastrichtian ceratopsids and hadrosaurs, two of the most abundant clades of dinosaurs in the Late Cretaceous of North America, is currently too poor to enable fair tests of endemicity and provincialism. In order to effectively test hypotheses regarding the causative mechanisms of biodiversity distribution, palaeontologists must demonstrate either that the fossil record preserves true biodiversity patterns at high levels of temporal resolution, or that methods exist that can adequately overcome sampling biases. The Western Interior region represents probably the most densely sampled Late Cretaceous terrestrial region worldwide [15,16], but even in this intensively sampled area, it is not currently possible to reconstruct diversity patterns at the regional scale. In order for palaeontologists to make a meaningful contribution to ecological hypotheses about future biodiversity change, we must focus our efforts on smaller scale case studies, where temporal resolution is high, stratigraphic correlation is well established, and where sampling biases are likely to be more homogeneous and can be more easily quantified. A good example of a recent such study is [65]. The results of multiple high-resolution case studies can then be compared globally to establish the rules that governed past biodiversity distributions.

Data accessibility. Raw data, additional methods, results, and figures can be found in the electronic supplementary material. Further raw data,

all code and a copy of the electronic supplementary material are available from the Dryad Digital Repository: https://doi.org/10.5061/dryad.bcc2fqzbz [66].

Author contributions. S.C.R.M. and R.J.B. designed the study. S.C.R.M., R.J.B., and C.D.D. wrote the manuscript. R.I.M. and S.C.R.M. collected data and generated supertrees. S.C.R.M., R.J.B., and C.D.D. ran analyses. All authors provided critical comments on the manuscript.

Competing interests. We declare we have no competing interests.

Funding. R.J.B. was funded by European Union's Horizon 2020 research and innovation programme under grant agreement no. 637483 during the course of this work.

Acknowledgements. Graeme Lloyd (University of Leeds) and David Button (Natural History Museum) wrote the original code to implement pBC. Caleb Brown (Royal Tyrrell Museum of Palaeontology) and an anonymous reviewer provided detailed and thoughtful comments that improved this manuscript. This is Paleobiology Database official publication 404.

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
