## [Peer Review File · Proceedings of the Royal Society B: Biological Sciences]

Review History

RSPB-2021-0692.R0 (Original submission)

Review form: Reviewer 1 (Caleb Brown)

Recommendation

Accept with minor revision (please list in comments)

Scientific importance: Is the manuscript an original and important contribution to its field?

Excellent

General interest: Is the paper of sufficient general interest?

Good

Quality of the paper: Is the overall quality of the paper suitable?

Good

Is the length of the paper justified?

Yes

Should the paper be seen by a specialist statistical reviewer?

Yes

Do you have any concerns about statistical analyses in this paper? If so, please specify them explicitly in your report.

No

It is a condition of publication that authors make their supporting data, code and materials available - either as supplementary material or hosted in an external repository. Please rate, if applicable, the supporting data on the following criteria.

Is it accessible?

Yes

Is it clear?

Yes

Is it adequate?

Yes

Do you have any ethical concerns with this paper?

No

Comments to the Author

See attached. (See Appendix A)

Review form: Reviewer 2

Recommendation

Major revision is needed (please make suggestions in comments)

Scientific importance: Is the manuscript an original and important contribution to its field?

Acceptable

General interest: Is the paper of sufficient general interest?

Good

Quality of the paper: Is the overall quality of the paper suitable?

Good

Is the length of the paper justified?

Yes

Should the paper be seen by a specialist statistical reviewer?

No

Do you have any concerns about statistical analyses in this paper? If so, please specify them explicitly in your report.

Yes

It is a condition of publication that authors make their supporting data, code and materials available - either as supplementary material or hosted in an external repository. Please rate, if applicable, the supporting data on the following criteria.

Is it accessible?

Yes

Is it clear?

Yes

Is it adequate?

Yes

Do you have any ethical concerns with this paper?

No

Comments to the Author

The present manuscript addresses the idea that dinosaur assemblages showed significant faunal provincialism in the western interior (Laramidia) during the late Cretaceous. The authors show that the raw dinosaur occurrence data show support for enhanced faunal provincialism between the north and south of Laramidia. However, application of methods meant to address the roles of sampling bias (generalized least squares), due to phenomena such as outcrop area, show that the result may in fact be the result of sampling bias. The authors therefore conclude that the data are not sufficient to test hypotheses relating to faunal provincialism in Laramidia during the late Cretaceous.

I cannot comment to any degree on how representative the dinosaur phylogeny is of current views, not being a dinosaur palaeobiologist.

Generally, however, the manuscript is well put together and I have no major issues with the methodologies as they follow what has been published in similar areas prior. I am, however, unconvinced of the novelty of the present study. The hypothesis at the heart of the manuscript has been tested quantitatively and published at least three previous times, with varying results. The present study differs only in using a different set of methodologies. Furthermore, according to the authors themselves a previous study also concluded that the pattern of significant provincialism may be a result of taphonomic bias (Chiarenza et al. [16]), as in the present study. That being said, I think that re-testing the same hypotheses in different ways is an important part of basic science.

Though the paper makes a recommendation that palaeobiologists focus on smaller spatiotemporal scales, it's unclear to me what such a study would look like (are there any areas that are currently well-sampled enough for these studies? How would studies at these smaller spatiotemporal scales address the "big picture" of the present manuscript (that the fossil record provides important context for understanding ongoing global change)? I think the novelty of the paper would be greatly enhanced by some clear, concise recommendations on this front.

Specific Comments:

Line 35-36 – "we must first demonstrate that actual biodiversity patterns are preserved in our reconstructions of past ecosystems" – The paper doesn't actually do this. The paper is starting from a (presumably) biased record. There is no way to know what the actual biodiversity pattern was. In this sense, this sentence is misrepresenting the paper. It is, true, but it does not represent the analyses conducted. This is the reason that several studies have started with the known record of terrestrial vertebrates and injected biases typical of the fossil record. This wording crops up again on line 101.

There are a number of spots where there aren't citations that I think there should be (see PDF)

Line 126-128 – It is unclear if the authors used bin_timepaleophy or just timepaleophy. The terrestrial fossil record tends to be dated using biostratigraphy or geological correlation, which means many sites and the fossils contained therein are often coarsely dated to stage level (or similar like North American Land Mammal Ages; demonstrated by the authors on line 129).

Timepaleophy assumes that dates are exact (in that they are direct dates) and `bin_timepaleophy` allows the potential date to vary within stages. As such, the binned method is typically preferred for palaeobiological trees.

Furthermore, it appears as if the authors have dated and used a single tree for each taxon (I could be missing it, if they did not). The exact FADs and LADs for most, if not all, of the taxa are not precisely known. It is therefore advantageous to construct a posterior distribution of trees and thus create a posterior distribution of metrics (pBC values). The posterior distribution of trees is created by allowing stochastic variation of ages within stages.

Randomization of data – the appropriate term is null model.

I was frustrated by the fact that the authors don't provide a model summary table in the body of the manuscript. I understand this is because the number of models and statistics is quite large. But surely they could present the top x number of models? That would greatly enhance readers' abilities to understand the analyses presented herein.

Figure 2 – it feels more natural to flip this plots so that latitude is on the x axis.

Figure 3 – It is unclear to me why this figure made it into the main body of the text.

Decision letter (RSPB-2021-0692.R0)

11-May-2021

Dear Dr Maidment:

Your manuscript has now been peer reviewed and the reviews have been assessed by an Associate Editor. The reviewers' comments (not including confidential comments to the Editor) and the comments from the Associate Editor are included at the end of this email for your reference. As you will see, the reviewers and the Editors have raised some concerns with your manuscript and we would like to invite you to revise your manuscript to address them.

Research ethics:

Use of animals and field studies:

It is a condition of publication that you make available the data and research materials supporting the results in the article. Please see our Data Sharing Policies (<https://royalsociety.org/journals/authors/author-guidelines/#data>). Datasets should be deposited in an appropriate publicly available repository and details of the associated accession number, link or DOI to the datasets must be included in the Data Accessibility section of the article (<https://royalsociety.org/journals/ethics-policies/data-sharing-mining/>). Reference(s) to datasets should also be included in the reference list of the article with DOIs (where available).

Please submit a copy of your revised paper within three weeks. If we do not hear from you within this time your manuscript will be rejected. If you are unable to meet this deadline please let us know as soon as possible, as we may be able to grant a short extension.

Best wishes,
Dr John Hutchinson, Editor
mailto: proceedingsb@royalsociety.org

Associate Editor
Board Member: 1
Comments to Author:

I am recommending revision of the manuscript. Both reviewers provide extensive comments on the manuscript, with the first recommending minor revisions and the second major revisions. I am not noting "major" revisions the second reviewers comments, and so am recommending the authors revise the manuscript as per the extensive comments provided by both reviewers. I find nothing amiss in the critiques provided and note that attention to these concerns will improve the manuscript.

Reviewer(s)' Comments to Author:
Referee: 1
Comments to the Author(s)
See attached

Referee: 2
Comments to the Author(s)

The present manuscript addresses the idea that dinosaur assemblages showed significant faunal provincialism in the western interior (Laramidia) during the late Cretaceous. The authors show that the raw dinosaur occurrence data show support for enhanced faunal provincialism between the north and south of Laramidia. However, application of methods meant to address the roles of sampling bias (generalized least squares), due to phenomena such as outcrop area, show that the result may in fact be the result of sampling bias. The authors therefore conclude that the data are not sufficient to test hypotheses relating to faunal provincialism in Laramidia during the late Cretaceous.

I cannot comment to any degree on how representative the dinosaur phylogeny is of current views, not being a dinosaur palaeobiologist.

Generally, however, the manuscript is well put together and I have no major issues with the methodologies as they follow what has been published in similar areas prior. I am, however, unconvinced of the novelty of the present study. The hypothesis at the heart of the manuscript has been tested quantitatively and published at least three previous times, with varying results. The present study differs only in using a different set of methodologies. Furthermore, according to the authors themselves a previous study also concluded that the pattern of significant provincialism may be a result of taphonomic bias (Chiarenza et al. [16]), as in the present study. That being said, I think that re-testing the same hypotheses in different ways is an important part of basic science.

Though the paper makes a recommendation that palaeobiologists focus on smaller spatiotemporal scales, it's unclear to me what such a study would look like (are there any areas that are currently well-sampled enough for these studies? How would studies at these smaller spatiotemporal scales address the "big picture" of the present manuscript (that the fossil record provides important context for understanding ongoing global change)? I think the novelty of the paper would be greatly enhanced by some clear, concise recommendations on this front.

Specific Comments:

Line 35-36 – “we must first demonstrate that actual biodiversity patterns are preserved in our reconstructions of past ecosystems” – The paper doesn’t actually do this. The paper is starting from a (presumably) biased record. There is no way to know what the actual biodiversity pattern was. In this sense, this sentence is misrepresenting the paper. It is, true, but it does not represent the analyses conducted. This is the reason that several studies have started with the known record of terrestrial vertebrates and injected biases typical of the fossil record. This wording crops up again on line 101.

There are a number of spots where there aren’t citations that I think there should be (see PDF)

Line 126-128 – It is unclear if the authors used bin_timepaleophy or just timepaleophy. The terrestrial fossil record tends to be dated using biostratigraphy or geological correlation, which means many sites and the fossils contained therein are often coarsely dated to stage level (or similar like North American Land Mammal Ages; demonstrated by the authors on line 129). Timepaleophy assumes that dates are exact (in that they are direct dates) and bin_timepaleophy allows the potential date to vary within stages. As such, the binned method is typically preferred for palaeobiological trees.

Furthermore, it appears as if the authors have dated and used a single tree for each taxon (I could be missing it, if they did not). The exact FADs and LADs for most, if not all, of the taxa are not precisely known. It is therefore advantageous to construct a posterior distribution of trees and thus create a posterior distribution of metrics (pBC values). The posterior distribution of trees is created by allowing stochastic variation of ages within stages.

Randomization of data – the appropriate term is null model.

I was frustrated by the fact that the authors don’t provide a model summary table in the body of the manuscript. I understand this is because the number of models and statistics is quite large. But surely they could present the top x number of models? That would greatly enhance readers’ abilities to understand the analyses presented herein.

Figure 2 – it feels more natural to flip this plots so that latitude is on the x axis.

Figure 3 – It is unclear to me why this figure made it into the main body of the text.

Author's Response to Decision Letter for (RSPB-2021-0692.R0)

See Appendix B.

Decision letter (RSPB-2021-0692.R1)

27-May-2021

Dear Dr Maidment

I am pleased to inform you that your manuscript entitled "Deep-time Biodiversity Patterns and the Dinosaurian Fossil Record of the Late Cretaceous Western Interior, North America" has been accepted for publication in Proceedings B. Congratulations!!

Data Accessibility section

Open Access

Your article has been estimated as being 8 pages long. Our Production Office will be able to confirm the exact length at proof stage.

Paper charges

Sincerely,

Dr John Hutchinson

Associate Editor:

Board Member

Comments to Author:

I have read through the revised manuscript, both the clean and marked copies, and read through the responses and rebuttals to comments made by the referees, and find that in my opinion, the authors have addressed those comments to my satisfaction. I am not recommending a second round of reviews as I believe the merit of the study presented by the authors is that it reveals points of contention around preservational bias and collecting biases and that true ecosystem-level/true diversity assessments of dinosaurs in the Late Cretaceous are not possible to detect from the current data.

Appendix A

Review of “Deep-time Biodiversity Patterns and the Dinosaurian Fossil Record of the Late Cretaceous Western Interior, USA”

Caleb Brown - April 12, 2021

The submission is a quantitative investigation into the discussion of north-south faunal provinciality/endemism in the dinosaur faunas of the Late Cretaceous of western North America. This is a debated topic in dinosaur paleobiology and the authors bring a new and much needed quantitative take on this issue. While this will certainly not be the last word in topic, this paper is a novel treatment of the data, with an informed consideration of the sampling biased inherent in palaeontology. The authors demonstrate an understanding and citation history of the existing literature on this topic, develop and execute a quantitative analysis that I believe is appropriate and uses justified methodologies, and the interpretations do not go beyond the results presented. The authors concentrate the analysis on the hadrosaurid and ceratopsid fossil record from the Campanian and Maastrichtian, and this is appropriate due to both high sampling, high diversity, good geographic coverage, and that the debate has centered on these.

The writing is clear, and the objectives of the paper are clearly defined and well laid out. I have itemized my suggested revisions below, and organized them in those that are more significant, those that are minor, typographical errors, and some that are purely suggestions. I do not believe that any of these suggested edits pose any real issue to the ultimate publication of the paper. I believe this will be a valuable contribution to the ongoing debate about dinosaur provinciality.

Major Issues:

Stage bins and diachronous formations:

Given that one of the main criticisms of some of the previous research supporting dinosaur provinciality has been that the formations being compared are diachronous, it is unfortunate that the temporal binning in the analysis was restricted to stage level. This will mean that the taxa that were separated by millions of years of time will be treated as contemporaneous in the test looking at whether or not they are clustered in space. This is similar to the issues pointed out by others in much of the work in dinosaur provinciality.

The authors realize this limitation, and mention it in the paper. The rationale for stage binning was that the dataset was “too sparse” to allow sampling below stage level. I am wondering if this needed to be explain more? Is the sample size too small (for taxa or occurrence)? Is the distribution of the taxa through time to skewed? Are the relevant data not recorded in the dataset? A look at the Campanian dataset show the vast majority of the occurrences are from the Late Campanian, but is this data just not stratigraphically controlled enough? What are the implications of the stage binning?

Sampling and pBC:

The authors nicely quantify the pattern of endemism in three clade/time bins, relative to that expected at random, and obtain strong and significant results. However, they then show that there is a strong correlation between occurrences data (used for the endemism analysis) and proxies for sampling, and suggest this pattern may be (in large part) due to unequal sampling. While I largely agree with this interpretation, I am left wondering if there is a more direct or formal test between the pBC and the sampling intensity? This almost feels like explaining away a positive result. Again, I largely agree with the interpretation, just wondering if there is a more direct test.

On a similar note, the sampling across latitudes in the Maastrichtian is much more even (relative to the Campanian), and no 'gap' is present, but the values for pBC for Maastrichtian ceratopsians are still more endemic than expected at random. The potential issues that sampling has on the pBC for the Campanian taxa are discussed in a fair amount of detail, but the weaker (but still strongly significant) yet potentially less biased Maastrichtian patterns are not really addressed. This may need to be expanded and explained in more detail. There is much more discussion of the results with the stronger sampling bias than there is those with the demonstrable weaker sampling bias.

Dinosaur Park Formation / 51°N:

There is some nuance to the Canadian dinosaur record that is not fully represented in the paper. The authors show a sampling peak at 51°N, and correlate this to the Dinosaur Park Formation. This is true, but this peak is not JUST the Dinosaur Park, but also would include the Oldman and, to a lesser extent, the Foremost formations, which both produce a diversity ceratopsid and hadrosaurs (though not as high as the DPF). For the southern peak at 36-37°N, several formations are implicated (Kirtland/Fruitland Formation, the Aguja Formation and the Kaiparowits Formation).

Perhaps more importantly, while a major outcrop of the Dinosaur Park Formation does outcrop at 51° N (i.e., Dinosaur Provincial Park - 80 km²), the Dinosaur Park Formation is much greater than this, and is exposed from 53° N down to 49° N. Within this range, the area from 51-49°N is actually sampled very well (especially 51° and 49°). For instance, in the Campanian, 49°N has the 2nd highest Ceratopsian occurrences (after 51°), and the 3rd highest hadrosaur occurrences (after 36° and 51°) (data from the paper). This pattern is also reflected in the DBCs, where 49° is third highest. The Oldman and Foremost also outcrop in the 51-49° range. Just as importantly, the dinosaur fauna within this 51-49° range is consistent across the latitudinal range. The dinosaurs in the Dinosaur Park Formation of Dinosaur Provincial Park (51°N), are the same as those from time equivalent beds (DPF or Oldman) from southern Alberta (e.g., Irvine, Onefour, Manyberries) (49°N). This pattern is seen throughout the Dinosaur Park and Oldman Formations, and had held up to good stratigraphically controlled sampling.

As a result, it is likely more accurate to refer to this 'northern' peaks as:

- Also due to the Oldman, and to a lesser extent, Foremost formations
- Either as a broader peak spanning from 51-49°, or two peaks (51° and 49°)

Due to the strong sampling at 49°N, I would argue the poor sampling interval in the Campanian does not start (i.e., northmost) at 50°N, but rather at 48°N.

Raw data:

I have done a quick review of the raw occurrence data. This review was biased toward the data that I am most familiar with, the Campanian Ceratopsian. There are several errors in the dataset (some larger than others), but it is unclear to what degree these will affect the analysis:

- The most obvious of these is the inclusion of *Triceratops horridus* within the Dinosaur Park Formation based on the Trenville Park specimen (TMP 1998.102.0001). This is incorrect, and both the Ryan and Russell 2001 publication (the PBDB citation) and the RTMP online database indicate that that this specimen is from the Scollard Fm. (which it is).
- Several (6?) *Monoclonius* occurrences are listed in the Dinosaur Park and Oldman formation. These specimens are either diagnosable *Centrosaurus apertus* specimens, or a non-diagnostic juveniles or subadult specimen.
- "*Monoclonius*" itself, is a 'wastebasket' taxon for non-diagnostic juvenile to subadult centrosaurine specimens. Its inclusion as a valid taxon will artificially skew the assemblages to having more taxic similarity. "*Monoclonius*" is included in the dataset from the Oldman and DPF of Alberta all the way down to the Cerro del Pueblo of Coahuila, (including occurrences in Montana and NM). This taxon does not show up on the phylogeny (as well as *Brachyceratops*), so many not be included in that analysis. If this is the case, it may be helpful to indicate which of the occurrences are excluded from the analysis.

I suspect these issues were just missed in an initial cross-checking of the occurrence data (or were knowingly excluded, but without mention in the paper). However, it makes sense that this type of occurrence based analysis is only as good as the raw data going in, so another check may be required. Similar issues may exist in the other datasets.

I assume that that the reported 'abundance values' in the datasets are not used in the analysis or factor into some proxy for sampling. If they are, these will need to be cross-checked as some of the numbers are gross over/under estimates.

PBDB bias:

The authors discuss many potential causes for the sampling gap between 51°N and 37°N including: amount of rock out crops area, fewer terrestrial rocks, lack of prospecting, sea level changes (common cause), historical collecting practices and landownership. To this I think it is worth considering another potential source of bias, unequal regional data entry into online databases (in this case the PBDB). It is odd that given the number of potential biased considered, there is no consideration for a potential bias in the online database. I can think of

several factors that may impact the likelihood of data being uploaded unequally to the PBDB (and result in geographic, stratigraphic biases):

- Edited popular books (e.g., the 2005 Dinosaur Park book) or monographs about specific sites/formations that include concretions of occurrence data represent 'low hanging fruit' and are likely to be incorporated into the PBDB earlier/easier than those records that are harder to track down, and/or spread out over a large number of publications.
- Museums that have searchable, online collections may be more likely to see their records added to the PBDB than museums that have collections records are not digitized or not publically accessible. And these museums will likely samples both geographically and stratigraphically different from each other.

I think it is at least worth a quick discussion that these potential biased between the collected fossil record and the record on the PBDB may exist.

Minor Issues:

Title and Line 44: Both in the title, and in the introduction (Line 44) the project is described regionally as in the western interior of the USA. However, the area of research testing dinosaur provinciality (both historically in the literature and in this analysis) spans both the USA and Canada (and to a lesser extent Mexico). For example, in the raw data used in this analysis, the Campanian Ceratopsian dataset (that showing the strongest pattern for provincially) is 62% Canadian occurrences, 32% USA, and 4% Mexico. The paper should more accurately represent this as the Western Interior of North America.

Line 51: I find is a bit odd that Lehman 1987 is not cited in this list. To me this paper is, in many ways, the origin of many of the ideas of dinosaur provinciality.

- Lehman, T.M. 1987. Late Maastrichtian paleoenvironments and dinosaur biogeography in the western interior of North America. *Palaeogeography, Palaeoclimatology, Palaeontology*, 60: 187-217.

Line 67: "Many studies advocating faunal endemism are based on taxonomic decisions that have proven controversial [e.g., 33]..." It is unclear if the author are suggesting that the taxonomy in the reference [33] is controversial, or that this reference suggests other taxonomy is controversial. As far as I know this reference does not question previous taxonomy, but adds new taxa (*Kosmoceratops* and *Utahceratops*). As far as I know there are no publications suggesting synonymy or other taxonomic corrections required for these taxa. Lucas et al., 2016 discuss the role of taxonomic decision in Chamosaurinae as effecting the idea of provinciality, but I do not see any specific questioning of the validity of *Kosmoceratops* or *Utahceratops*. This may need to be rephrased.

Lines 72-86: The discussion of the three previous attempts to quantitatively test the idea of Late Cretaceous dinosaur provinciality is out of order. In my mind these should be presented chronologically, so as to show the progression in scientific thinking and ideas through time.

Instead this research is reviewed in an anti-chronological order, making the progression of ideas difficult to see.

Line 99: It would be useful to quickly define “collection” vs. “occurrence”

Line 108: “We resolved polytomies by removing *Nedoceratops*, a taxon recently found to be invalid,...” I would suggest changing “found to be” to “suggested to be” as there is not a consensus among ceratopsian workers regarding this synonymy. A reference should also be added to support this referral of it is invalid.

- Reference suggesting invalid: Scannella, J.B., and Horner, J.R. 2011. ‘*Nedoceratops*’: an example of a transitional morphology. PLoS ONE, 6: e28705.
- Reference suggesting valid: Farke, A.A. 2011. Anatomy and taxonomic status of the chasmosaurine ceratopsid *Nedoceratops hatcheri* from the Upper Cretaceous Lance Formation of Wyoming, USA. PLoS ONE, 6: e16196.

Line 171: It should be clarified if modern latitude or palaeolatitude is being used the bin these occurrences. After reading thought I suspect it is the former, but best to clarify.

Figure 1D: What do the error bars in Figure 1D represent? Are they 95% CI? Range of all values obtaining in the subsampling?

Figure 1D: The error bars of Figure 1D are drawn such that when there is overlap, the Maastrichtian bar obscures the Campanian bar making the amount of overlap difficult to see. Either that or there are no error bars for the Campanian < 15.

Figure 2: The acronym DBC is used in the legend but not defined. I assume this means Dinosaur Bearing collections. This should be included in the figure caption or elsewhere.

Typographical issues:

Line 51: Numerous workers have argued for strong provinciality IN Laramida

Line 54 (and elsewhere): I find the capitalization of Northern and Southern Larimidia confusing. Are these defined geopolitical regions requiring capitalization? I have always seen these as referring to the northern and southern portions of Laramida. I am honestly not sure, but this this seems a little odd. Other works do not capitalize these, E.g.,:

- Ryan, M.J., Evans, D.C., Currie, P.J. and Loewen, M.A., 2014. A new chasmosaurine from **northern** Laramida expands frill disparity in ceratopsid dinosaurs. *Naturwissenschaften*, 101(6), pp.505-512.
- Sampson, S.D., Lund, E.K., Loewen, M.A., Farke, A.A. and Clayton, K.E., 2013. A remarkable short-snouted horned dinosaur from the Late Cretaceous (late Campanian) of **southern** Laramida. *Proceedings of the Royal Society B: Biological Sciences*, 280(1766), p.20131186.

Line 401: “sympatry”

Pure suggestions:

Figure 1: For easy of visualization, it may be worth changing one set of the symbols from a circle to a square, triangle, or diamond in Figure 1D. This way colour is not the only differentiating factor, but shape as well. This would help for people with limited color vision and greyscale printing.

Figure 2: Add an equidimensional, latitude aligned map (as in Figure 3) along the right-hand side of the plots in Figure 2 (may need to compress the plots). This would allow for a quicker and easier reference to where these sampling and diversity peaks are located, rather than having to compare the latitude to a map. This map could be plain (just political boundaries) or could indicate strata density (as in Fig 3A), but for both the Campanian and Maastrichtian. Purely a suggestion.

Appendix B

Responses to reviewer comments

Reviewer(s)' Comments to Author:

Referee: 1

Comments to the Author(s)

Caleb Brown - April 12, 2021

The submission is a quantitative investigation into the discussion of north-south faunal provinciality/endemism in the dinosaur faunas of the Late Cretaceous of western North America. This is a debated topic in dinosaur paleobiology and the authors bring a new and much needed quantitative take on this issue. While this will certainly not be the last word in topic, this paper is a novel treatment of the data, with an informed consideration of the sampling biased inherent in palaeontology. The authors demonstrate an understanding and citation history of the existing literature on this topic, develop and execute a quantitative analysis that I believe is appropriate and uses justified methodologies, and the interpretations do not go beyond the results presented. The authors concentrate the analysis on the hadrosaurid and ceratopsid fossil record from the Campanian and Maastrichtian, and this is appropriate due to both high sampling, high diversity, good geographic coverage, and that the debate has centered on these.

The writing is clear, and the objectives of the paper are clearly defined and well laid out. I have itemized my suggested revisions below, and organized them in those that are more significant, those that are minor, typographical errors, and some that are purely suggestions. I do not believe that any of these suggested edits pose any real issue to the ultimate publication of the paper. I believe this will be a valuable contribution to the ongoing debate about dinosaur provinciality.

Major Issues:

Stage bins and diachronous formations:

Given that one of the main criticisms of some of the previous research supporting dinosaur provinciality has been that the formations being compared are diachronous, it is unfortunate that the temporal binning in the analysis was restricted to stage level. This will mean that the taxa that were separated by millions of years of time will be treated as contemporaneous in the test looking at whether or not they are clustered in space. This is similar to the issues pointed out by others in much of the work in dinosaur provinciality. The authors realize this limitation, and mention it in the paper. The rationale for stage binning was that the dataset was “too sparse” to allow sampling below stage level. I am wondering if this needed to be explain more? Is the sample size too small (for taxa or occurrence)? Is the distribution of the taxa through time to skewed? Are the relevant data not recorded in the dataset? A look at the Campanian dataset show the vast majority of the occurrences are from the Late Campanian, but is this data just not stratigraphically controlled enough? What are the implications of the stage binning?

The main issue is that too few taxa would be present in each time bin to calculate pBC if we divided it ideally (for example, into the time bins suggested for this area in Dean et al. 2020 ref 17). But also, the aim of this paper is really to test pre-existing hypotheses about faunal endemism, which have

always been presented at stage level. Thus, in order to compare with previous studies, and also to allow us to have enough data in each time bin to carry out statistical tests on occurrence data (for example), we stuck with Stage level bins. The paragraph discussing this has been nuanced to read “Whilst it would be ideal to use high-resolution bins to test patterns of biogeography through the Late Cretaceous [17], too few taxa would be present in each bin to permit the use of pBC. Consequently, we divided taxa into Campanian and Maastrichtian time bins, which also has the benefit of allowing for comparison between previous studies of faunal provincialism in this area.”

Sampling and pBC:

The authors nicely quantify the pattern of endemism in three clade/time bins, relative to that expected at random, and obtain strong and significant results. However, they then show that there is a strong correlation between occurrences data (used for the endemism analysis) and proxies for sampling, and suggest this pattern may be (in large part) due to unequal sampling. While I largely agree with this interpretation, I am left wondering if there is a more direct or formal test between the pBC and the sampling intensity? This almost feels like explaining away a positive result. Again, I largely agree with the interpretation, just wondering if there is a more direct test.

This is an excellent point and one that we have also thought about since obtaining the results of the analyses we carried out here. pBC as currently implemented does not control for sampling bias, and it would certainly be interesting to develop the method further to build in a way of accounting for sampling. However, that is a major methodological development and well outside the scope of this paper.

We have now added a secondary theoretical approach to support our findings. We ran a second pBC test where we randomly removed 95% of ceratopsian taxa from the Maastrichtian that occurred between 35 and 50 degrees of latitude. We chose these latitudinal boundaries to enforce a similar bimodal latitudinal diversity gradient as seen in the Campanian. The remaining distribution of species was used to re-run pBC analyses (with a μ of 10), and this process was repeated 1000 times for increased accuracy of results. Forcing a Campanian-like sampling pattern in the Maastrichtian substantially reduced the probability of lower pBC scores for Maastrichtian ceratopsians, indicating that it is likely that the pBC scores for the Campanian are caused by sampling bias. This methodology and associated results have now been included within the manuscript and OSM, and the full script to run this analysis is also provided.

On a similar note, the sampling across latitudes in the Maastrichtian is much more even (relative to the Campanian), and no ‘gap’ is present, but the values for pBC for Maastrichtian ceratopsians are still more endemic than expected at random. The potential issues that sampling has on the pBC for the Campanian taxa are discussed in a fair amount of detail, but the weaker (but still strongly significant) yet potentially less biased Maastrichtian patterns are not really addressed. This may need to be expanded and explained in more detail. There is much more discussion of the results with the stronger sampling bias than there is those with the demonstrable weaker sampling bias.

We think the reviewer has slightly misunderstood the results here. It’s important to say that the Maastrichtian record is very biased by sampling; indeed, the statistical tests reported in Fig. 2 demonstrate that the correlation between occurrences and collections is actually slightly stronger for Maastrichtian ceratopsids than the Campanian ceratopsids, while for hadrosaurs, the strength of correlations are virtually the same. The difference between the Campanian and Maastrichtian is that in the Campanian, sampling is concentrated in two latitudinal ‘peaks’, while in the Maastrichtian it is more evenly spread over a broad latitudinal range, and our interpretation is that this is why the

Campanian record is found to be more 'endemic' than the Maastrichtian. It does not suggest (contra to the reviewer's statement) that the Maastrichtian is more 'weakly' biased by sampling. Our discussion is focused on the Campanian because this is the time slice where previous studies have focused discussions about faunal endemicity, and fundamentally this study is about testing those previous hypotheses and, to a lesser extent, we are limited by the length of the paper. Our comments about sources of sampling bias are generally equally applicable to the Maastrichtian, although it would appear that in the Maastrichtian, there has been a more even sampling effort across the latitudinal range of the Western Interior.

Dinosaur Park Formation / 51°N:

There is some nuance to the Canadian dinosaur record that is not fully represented in the paper. The authors show a sampling peak at 51°N, and correlate this to the Dinosaur Park Formation. This is true, but this peak is not JUST the Dinosaur Park, but also would include the Oldman and, to a lesser extent, the Foremost formations, which both produce a diversity ceratopsid and hadrosaurs (though not as high as the DPF). For the southern peak at 36-37°N, several formations are implicated (Kirtland/Fruitland Formation, the Aguja Formation and the Kaiparowits Formation).

Perhaps more importantly, while a major outcrop of the Dinosaur Park Formation does outcrop at 51° N (i.e., Dinosaur Provincial Park - 80 km²), the Dinosaur Park Formation is much greater than this, and is exposed from 53° N down to 49° N. Within this range, the area from 51-49°N is actually sampled very well (especially 51° and 49°). For instance, in the Campanian, 49°N has the 2nd highest Ceratopsian occurrences (after 51°), and the 3rd highest hadrosaur occurrences (after 36° and 51°) (data from the paper). This pattern is also reflected in the DBCs, where 49° is third highest. The Oldman and Foremost also outcrop in the 51-49° range. Just as importantly, the dinosaur fauna within this 51-49° range is consistent across the latitudinal range. The dinosaurs in the Dinosaur Park Formation of Dinosaur Provincial Park (51°N), are the same as those from time equivalent beds (DPF or Oldman) from southern Alberta (e.g., Irvine, Onefour, Manyberries) (49°N). This pattern is seen throughout the Dinosaur Park and Oldman Formations, and had held up to good stratigraphically controlled sampling.

As a result, it is likely more accurate to refer to this 'northern' peaks as:

- Also due to the Oldman, and to a lesser extent, Foremost formations
- Either as a broader peak spanning from 51-49°, or two peaks (51° and 49°)

Due to the strong sampling at 49°N, I would argue the poor sampling interval in the Campanian does not start (i.e., northmost) at 50°N, but rather at 48°N.

Yes absolutely. This has been changed as suggested above throughout the text.

Raw data:

I have done a quick review of the raw occurrence data. This review was biased toward the data that I am most familiar with, the Campanian Ceratopsian. There are several errors in the dataset (some larger than others), but it is unclear to what degree these will affect the analysis:

- The most obvious of these is the inclusion of *Triceratops horridus* within the Dinosaur Park Formation based on the Trenville Park specimen (TMP 1998.102.0001). This is incorrect, and both the Ryan and Russell 2001 publication (the PBDB citation) and the RTMP online database indicate that that this specimen is from the Scollard Fm. (which it is).
- Several (6?) *Monoclonius* occurrences are listed in the Dinosaur Park and Oldman formation. These specimen are either diagnosable *Centrosaurus apertus* specimens,

or a non-diagnostic juveniles or subadult specimen.

- "*Monoclonius*" itself, is a 'wastebasket' taxon for non-diagnostic juvenile to subadult centrosaurine specimens. Its inclusion as a valid taxon will artificially skew the assemblages to having more taxic similarity. "*Monoclonius*" is included in the dataset from the Oldman and DPF of Alberta all the way down to the Cerro del Pueblo of Coahuila, (including occurrences in Montana and NM). This taxon does not show up on the phylogeny (as well as *Brachyceratops*), so many not be included in that analysis. If this is the case, it may be helpful to indicate which of the occurrences are excluded from the analysis.

I suspect these issues were just missed in an initial cross-checking of the occurrence data (or were knowingly excluded, but without mention in the paper). However, it makes sense that this type of occurrence based analysis is only as good as the raw data going in, so another check may be required. Similar issues may exist in the other datasets.

The data used for pBC analyses (and the phylogenetic trees) and the occurrence data used for the sampling bias analyses are two different data sets.

pBC uses species and temporal ranges. We only included currently accepted species into the phylogeny (although this is somewhat tricky for ceratopsids because there is little agreement among workers about what is valid and what is not, so the tree that we produced is what we feel is a fair representation of a consensus view). Thus only currently accepted species are used, and as the reviewer notes, "*Monoclonius*" and "*Brachyceratops*" are not included as they are not accepted as valid by the majority of authors. The stratigraphic ages of taxa were taken from the literature and sources are provided in the supplementary data. PBDB data was not the primary source of information for pBC analyses.

Conversely, occurrence data is simply an occurrence of a ceratopsid fossil. It doesn't make any difference what taxon that fossil is, as long as we can agree that it is a ceratopsid of some sort. So indeterminate or dubiously named taxa can be included because we agree that they are ceratopsids *of some sort*. Thus it was not necessary for us to update the occurrence list for currently accepted names.

We also noted the presence of a "*Triceratops*" specimen out of place in the data, but assumed it was mis-identified and, as such, included it as a ceratopsid of some sort. We thank the reviewer for bringing to our attention that it should be in the Maastrichtian Scollard Fm rather than in the Campanian. However, the removal of one occurrence will make absolutely no difference whatsoever to the results we have obtained or the patterns we have observed.

I assume that that the reported 'abundance values' in the datasets are not used in the analysis or factor into some proxy for sampling. If they are, these will need to be cross-checked as some of the numbers are gross over/under estimates.

Yes this is correct; abundance values from the PBDB are not used and we agree they are quite unreliable.

PBDB bias:

The authors discuss many potential causes for the sampling gap between 51°N and 37°N including: amount of rock out crops area, fewer terrestrial rocks, lack of prospecting, sea level changes (common cause), historical collecting practices and landownership. To this I think it is

worth considering another potential source of bias, unequal regional data entry into online databases (in this case the PBDB). It is odd that given the number of potential biased considered, there is no consideration for a potential bias in the online database. I can think of several factors that may impact the likelihood of data being uploaded unequally to the PBDB (and result in geographic, stratigraphic biases):

- Edited popular books (e.g., the 2005 Dinosaur Park book) or monographs about specific sites/formations that include concretions of occurrence data represent 'low hanging fruit' and are likely to be incorporated into the PBDB earlier/easier than those records that are harder to track down, and/or spread out over a large number of publications.
- Museums that have searchable, online collections may be more likely to see their records added to the PBDB than museums that have collections records are not digitized or not publically accessible. And these museums will likely samples both geographically and stratigraphically different from each other.

I think it is at least worth a quick discussion that these potential biased between the collected fossil record and the record on the PBDB may exist.

Good point. We've added a discussion on this.

Minor Issues:

Title and Line 44: Both in the title, and in the introduction (Line 44) the project is described regionally as in the western interior of the USA. However, the area of research testing dinosaur provinciality (both historically in the literature and in this analysis) spans both the USA and Canada (and to a lesser extent Mexico). For example, in the raw data used in this analysis, the Campanian Ceratopsian dataset (that showing the strongest pattern for provincially) is 62% Canadian occurrences, 32% USA, and 4% Mexico. The paper should more accurately represent this as the Western Interior of North America.

Changed to USA to North America

Line 51: I find is a bit odd that Lehman 1987 is not cited in this list. To me this paper is, in many ways, the origin of many of the ideas of dinosaur provinciality.

- Lehman, T.M. 1987. Late Maastrichtian paleoenvironments and dinosaur biogeography in the western interior of North America. *Palaeogeography, Palaeoclimatology, Palaeontology*, 60: 187-217.

Citation added as suggested

Line 67: "Many studies advocating faunal endemism are based on taxonomic decisions that have proven controversial [e.g., 33]..." It is unclear if the author are suggesting that the taxonomy in the reference [33] is controversial, or that this reference suggests other taxonomy is controversial. As far as I know this reference does not question previous taxonomy, but adds new taxa (*Kosmoceratops* and *Utahceratops*). As far as I know there are no publications suggesting synonymy or other taxonomic corrections required for these taxa. Lucas et al., 2016 discuss the role of taxonomic decision in Chamosaurinae as effecting the idea of provinciality, but I do not see any specific questioning of the validity of *Kosmoceratops* or *Utahceratops*. This may need to be rephrased.

The point that we were trying to make here was that in ref 33, *Kosmoceratops* was named and identified as only being found in southern Laramidia. But in ref 38, *Kosmoceratops* was identified in northern Laramidia (also discussed in 36). We agree with the reviewer however that this was poorly referenced and so have removed the reference to 33 in line 67.

Lines 72-86: The discussion of the three previous attempts to quantitatively test the idea of Late Cretaceous dinosaur provinciality is out of order. In my mind these should be presented chronologically, so as to show the progression in scientific thinking and ideas through time. Instead this research is reviewed in an anti-chronological order, making the progression of ideas difficult to see.

We swapped the discussion of Gates and Berry so that Gates, published first, is discussed before Berry, which was published later. Vavrek and Larsson is left to last because that discusses the Maastrichtian, while the other two discuss the Campanian (which is really the focus of this paper).

Line 99: It would be useful to quickly define “collection” vs. “occurrence”

Added

Line 108: “We resolved polytomies by removing *Nedoceratops*, a taxon recently found to be invalid,…” I would suggest changing “found to be” to “suggested to be” as there is not a consensus among ceratopsian workers regarding this synonymy. A reference should also be added to support this referral of it is invalid.

- Reference suggesting invalid: Scannella, J.B., and Horner, J.R. 2011. ‘*Nedoceratops*’: an example of a transitional morphology. PLoS ONE, 6: e28705.

- Reference suggesting valid: Farke, A.A. 2011. Anatomy and taxonomic status of the chasmosaurine ceratopsid *Nedoceratops hatcheri* from the Upper Cretaceous Lance Formation of Wyoming, USA. PLoS ONE, 6: e16196.

We changed the wording to “a taxon some workers consider to be invalid” and added references.

Line 171: It should be clarified if modern latitude or palaeolatitude is being used the bin these occurrences. After reading thought I suspect it is the former, but best to clarify.

Clarified by adding “(latitude is modern latitude, rather than palaeolatitude)”

Figure 1D: What do the error bars in Figure 1D represent? Are they 95% CI? Range of all values obtaining in the subsampling?

They represent 95% confidence intervals – we have added this to the figure caption

Figure 1D: The error bars of Figure 1D are drawn such that when there is overlap, the Maastrichtian bar obscures the Campanian bar making the amount of overlap difficult to see. Either that or there are no error bars for the Campanian < 15.

I have adjusted the figure so that the error bars are no longer lying on top of each other.

Figure 2: The acronym DBC is used in the legend but not defined. I assume this means Dinosaur Bearing collections. This should be included in the figure caption or elsewhere.

This is now defined in the figure caption

Typographical issues:

Line 51: Numerous workers have argued for strong provinciality IN Laramida

Corrected

Line 54 (and elsewhere): I find the capitalization of Northern and Southern Larimidia confusing. Are these defined geopolitical regions requiring capitalization? I have always seen these as referring to the northern and southern portions of Laramidia. I am honestly not sure, but this this seems a little odd. Other works do not capitalize these, E.g.,:

- Ryan, M.J., Evans, D.C., Currie, P.J. and Loewen, M.A., 2014. A new chasmosaurine from **northern** Laramidia expands frill disparity in ceratopsid dinosaurs. *Naturwissenschaften*, 101(6), pp.505-512.

- Sampson, S.D., Lund, E.K., Loewen, M.A., Farke, A.A. and Clayton, K.E., 2013. A remarkable short-snouted horned dinosaur from the Late Cretaceous (late Campanian) of **southern** Laramidia. *Proceedings of the Royal Society B: Biological Sciences*, 280(1766), p.20131186.

I wasn't sure either! They are now uncapitalized throughout.

Line 401: "sympatry"

Corrected.

Pure suggestions:

Figure 1: For easy of visualization, it may be worth changing one set of the symbols from a circle to a square, triangle, or diamond in Figure 1D. This way colour is not the only differentiating factor, but shape as well. This would help for people with limited color vision and greyscale printing.

Good idea – done.

Figure 2: Add an equidimensional, latitude aligned map (as in Figure 3) along the right-hand side of the plots in Figure 2 (may need to compress the plots). This would allow for a quicker and easier reference to where these sampling and diversity peaks are located, rather than having to compare the latitude to a map. This map could be plain (just political boundaries) or could indicate strata density (as in Fig 3A), but for both the Campanian and Maastrichtian. Purely a suggestion.

This would certainly be nice to have but we feel it will squash the plots so much that the data will no longer be visible.

Referee: 2

Comments to the Author(s)

The present manuscript addresses the idea that dinosaur assemblages showed significant faunal provincialism in the western interior (Laramidia) during the late Cretaceous. The authors show that the raw dinosaur occurrence data show support for enhanced faunal provincialism between the north and south of Laramidia. However, application of methods meant to address the roles of sampling bias (generalized least squares), due to phenomena such as outcrop area, show that the result may in fact be the result of sampling bias. The authors therefore conclude that the data are not sufficient to test hypotheses relating to faunal provincialism in Laramidia during the late

Cretaceous.

I cannot comment to any degree on how representative the dinosaur phylogeny is of current views, not being a dinosaur palaeobiologist.

Generally, however, the manuscript is well put together and I have no major issues with the methodologies as they follow what has been published in similar areas prior. I am, however, unconvinced of the novelty of the present study. The hypothesis at the heart of the manuscript has been tested quantitatively and published at least three previous times, with varying results. The present study differs only in using a different set of methodologies. Furthermore, according to the authors themselves a previous study also concluded that the pattern of significant provincialism may be a result of taphonomic bias (Chiarenza et al. [16]), as in the present study. That being said, I think that re-testing the same hypotheses in different ways is an important part of basic science.

Only two studies have investigated endemism in the Campanian, and one presented inconclusive results with respect to dinosaurs, specifically suggesting further investigation into the causes of dinosaur distribution in the Western Interior during the Campanian. Chiarenza et al. did not specifically test hypotheses of faunal endemism; their paper was on ecological niche modelling. However, they did discuss the implications of their work for such hypotheses.

Though the paper makes a recommendation that palaeobiologists focus on smaller spatiotemporal scales, it's unclear to me what such a study would look like (are there any areas that are currently well-sampled enough for these studies? How would studies at these smaller spatiotemporal scales address the "big picture" of the present manuscript (that the fossil record provides important context for understanding ongoing global change)? I think the novelty of the paper would be greatly enhanced by some clear, concise recommendations on this front.

We end the paper with: "In order for palaeontologists to make a meaningful contribution to ecological hypotheses about future biodiversity change, we must focus our efforts on smaller-scale case studies, where temporal resolution is high, stratigraphic correlation is well-established, and where sampling biases are likely to be more homogenous and can be more easily quantified. The results of multiple high-resolution case studies can then be compared globally to establish the rules that governed past biodiversity distributions." These are our recommendations, but to address the reviewer's point we additionally cite a study which does the sorts of things we recommend.

Specific Comments:

Line 35-36 – "we must first demonstrate that actual biodiversity patterns are preserved in our reconstructions of past ecosystems" – The paper doesn't actually do this. The paper is starting from a (presumably) biased record. There is no way to know what the actual biodiversity pattern was. In this sense, this sentence is misrepresenting the paper. It is, true, but it does not represent the analyses conducted. This is the reason that several studies have started with the known record of terrestrial vertebrates and injected biases typical of the fossil record. This wording crops up again on line 101.

Yes, that's the point of the paper. Palaeontologists ("we") must demonstrate this, and the message from this paper is that at the moment, with the data that we have, we can't.

There are a number of spots where there aren't citations that I think there should be (see PDF)

Only one pdf was attached to the reviewer comments we received and that was the one from reviewer 1.

Line 126-128 – It is unclear if the authors used bin_timepaleophy or just timepaleophy. The terrestrial fossil record tends to be dated using biostratigraphy or geological correlation, which means many sites and the fossils contained therein are often coarsely dated to stage level (or similar like North American Land Mammal Ages; demonstrated by the authors on line 129). Timepaleophy assumes that dates are exact (in that they are direct dates) and bin_timepaleophy allows the potential date to vary within stages. As such, the binned method is typically preferred for palaeobiological trees.

Furthermore, it appears as if the authors have dated and used a single tree for each taxon (I could be missing it, if they did not). The exact FADs and LADs for most, if not all, of the taxa are not precisely known. It is therefore advantageous to construct a posterior distribution of trees and thus create a posterior distribution of metrics (pBC values). The posterior distribution of trees is created by allowing stochastic variation of ages within stages.

We understand and agree with the reviewer's general point, but we don't think it is relevant to our study. Although the range of time between the FAD and LAD for our taxa almost always represents uncertainty and is based on dating of formations in which the taxa are found, dating of formations is much more accurate than stage-level, as can be seen in the supplementary data. For example, the Dinosaur Park Formation is dated as 76.5-74.8 Ma. Whilst allowing the dating of taxa to vary within the range of uncertainty using the bin_timepaleophyl function would be necessary at higher levels of temporal resolution, in this case it would make no difference because the uncertainty ranges are much less than the length of time represented by our stage-level bins. Thus, there does not seem to be any value in producing a posterior distribution of trees and pBC values.

Randomization of data – the appropriate term is null model.

Added

I was frustrated by the fact that the authors don't provide a model summary table in the body of the manuscript. I understand this is because the number of models and statistics is quite large. But surely they could present the top x number of models? That would greatly enhance readers' abilities to understand the analyses presented herein.

As an additional point here, we realised that some minor errors existed within our GLS results. For instance, MGVF was masked by "combined outcrop", yet was being used in GLS models against "total outcrop". We also updated the masking approach to also mask for locations which contained both outcrop and tetrapod occurrences (sourced from the PBDB) to more accurately reflect which conditions were impacting hadrosaur and ceratopsian occurrences specifically. As such, we generated new values for use in GLS results and re-ran them to obtain new results. These results are provided in OSM, and the main text has been updated to reflect these results. Changes to results are minor, and do not impact the main conclusions of this work.

There are space constraints on what we are able to include in the main text – even presenting the top 3 models for each analysis (including both old masking and updated masking processes) will require a table with 24 model results that will take up substantial space. In general, as noted in the main text, very few of the models have significant variables, and based on our updated results most of the top models (based on AICc weights) are null models. Given this, it seems preferential to report

all the models simultaneously in the supplement rather than arbitrarily pulling a certain number of them out into the main text.

Figure 2 – it feels more natural to flip this plots so that latitude is on the x axis.

Having latitude on the x-axis is counterintuitive to us. Latitude is north-south variation, so displaying it on the y-axis makes more sense in a visual way

Figure 3 – It is unclear to me why this figure made it into the main body of the text.

This figure was removed to the supplementary data.